# Recent Trends in the Petasis Reaction: A Review of Novel Catalytic Synthetic Approaches with Applications of the Petasis Reaction

**DOI:** 10.3390/molecules28248032

**Published:** 2023-12-10

**Authors:** Sadaf Saeed, Saba Munawar, Sajjad Ahmad, Asim Mansha, Ameer Fawad Zahoor, Ali Irfan, Ahmad Irfan, Katarzyna Kotwica-Mojzych, Malgorzata Soroka, Mariola Głowacka, Mariusz Mojzych

**Affiliations:** 1Medicinal Chemistry Research Lab, Department of Chemistry, Government College University Faisalabad, Faisalabad 38000, Pakistan; sadafsaeed932@gmail.com (S.S.); sabamunawar597@gmail.com (S.M.); asimmansha@gcuf.edu.pk (A.M.); raialiirfan@gmail.com (A.I.); 2Department of Basic Sciences and Humanities, University of Engineering and Technology Lahore, Faisalabad Campus, Faisalabad 38000, Pakistan; sajjad.ahmad@uet.edu.pk; 3Department of Chemistry, College of Science, King Khalid University, Abha 61413, Saudi Arabia; ahmeed@kku.edu.sa; 4Department of Histology, Embryology and Cytophysiology of the Department of Basic Sciences, Medical University of Lublin, Radziwiłłowska 11, 20-080 Lublin, Poland; katarzynakotwicamojzych@umlub.pl; 5Faculty of Medicine, Collegium Medicum, The Mazovian Academy in Plock, Pl. Dąbrowskiego 2, 09-402 Płock, Poland; m.soroka@mazowiecka.edu.pl; 6Faculty of Health Sciences, Collegium Medicum, The Mazovian Academy in Plock, Pl. Dąbrowskiego 2, 09-402 Płock, Poland; m.glowacka@mazowiecka.edu.pl; 7Department of Chemistry, Siedlce University of Natural Sciences and Humanities, 3-go Maja 54, 08-110 Siedlce, Poland

**Keywords:** Petasis reaction, multicomponent reactions, boronic acids, transition metals, BINOL catalyst

## Abstract

The Petasis reaction, also called the Petasis Borono–Mannich reaction, is a multicomponent reaction that couples a carbonyl derivative, an amine and boronic acids to yield substituted amines. The reaction proceeds efficiently in the presence or absence of a specific catalyst and solvent. By employing this reaction, a diverse range of chiral derivatives can easily be obtained, including α-amino acids. A broad substrate scope, high yields, distinct functional group tolerance and the availability of diverse catalytic systems constitute key features of this reaction. In this review article, attention has been drawn toward the recently reported methodologies for executing the Petasis reaction to produce structurally simple to complex aryl/allyl amino scaffolds.

## 1. Introduction

Multicomponent reactions constitute the coupling between more than two components which are incorporated into the final product [1]. A number of advantages are accredited to this type of reaction. These include atom economy, as the majority of reactants become part of the product; efficiency, because of high product yield; convergence; and exhibiting a high index of bond formation. Some of the important multicomponent reactions are Ugi, Strecker, Passerini, Hantzch, Betti and Biginelli reactions [2,3,4,5,6]. Multicomponent reactions find their applications in the stereoselective transformations including the synthesis/reactions of heterocycles such as pyrazines, epoxides and aziridines [7,8,9,10]. These reactions have been widely employed in medicinal and combinatorial chemistry for purposes of the drug-discovery process to construct biologically active scaffolds [11].

The Petasis reaction is one of the significant multicomponent reactions [12]. It takes place between an amine, a carbonyl group and boronic acids. This reaction provides highly stereoselective products under facile reaction conditions (short duration and mild temperature), providing access to a wide range of substrates. It can proceed either via nanoparticles, Lewis acids, metal catalysts or catalyst-free conditions with minimal requirement of protecting groups [13,14,15]. It was first reported by Petasis and his coworkers in 1993, who obtained allyl amines, i.e., naftifine, an anti-fungal agent from paraformaldehyde, secondary amine and *E*-vinyl boronic acid, constituting an archetype of the Petasis reaction [16]. The other transformations of the Petasis reaction involved Petasis–olefination, which is the dimethyl titanocene-mediated olefination of ketone or aldehyde and the Petasis–Ferrier reaction which is the formation of tetrahydrofuran from cyclic acetals. It has evolved into a highly functionalized multicomponent reaction, with a lot of key features. A general reaction scheme of Petasis reaction is presented in Figure 1 as follows [17]:

Focusing on the applications of this reaction, a variety of lead compounds with significant biological potential have been constructed, such as naftifine, which is an active antifungal compound. In addition to this, the synthesis of (−)-clavosolide A **1**, (−)-okilactomycin **2**, (−)-kendomycin **3** and (+)-phorboxazole A **4** was also accomplished via this reaction. The Petasis–Ferrier reaction played an important role in the total synthesis of (−)-clavosolide A **1** [18]. The (−)-okilactomycin **2** has been reported to be a significant cytotoxic compound with an IC_50_ value of 0.037 and 0.09 µg/mL against lymphoid leukemia and cell line P388, respectively [19]. The Petasis–Ferrier reaction is one of the many steps in the total synthesis of (−)-kendomycin **3** and (+)-phorboxazole A **4** [20,21]. Figure 2 contains the structures of the above-mentioned compounds.

The Petasis reaction provides a powerful route to access tertiary amines, amino esters, allenes, amino acids, functionalized γ-lactams, 2-aminothiophenes and thienodiazepines, and many reports regarding this reaction have been published during the last few years [22,23,24,25,26,27,28]. The contents of this review have been divided based on different conditions employed for Petasis reactions (such as Petasis reactions catalyzed by nanoparticles, chiral catalysts, acid-catalyzed reactions, catalyst-free reactions, solvent-free reactions, base-catalyzed reactions and miscellaneous reactions) reported in recent years.

## 2. Review of Literature

### 2.1. Petasis Reaction Catalyzed by Nanoparticles

Alkylaminophenols are a part of different pharmacological drugs, e.g., amodiaquine and topotecan [29,30]. Although the first synthesis of these compounds was catalyst-free, affording a low yield range [31], keeping those considerations in mind, Kulkarni et al. reported in 2015 for the first time, the catalyst-based Petasis borono–Mannich reaction to achieve alkylaminophenols (72–93%). They examined the effect of spinel ferrites as catalysts (Fe_3_O_4_, NiFe_2_O_4_, CoFe_2_O_4_ and CuFe_2_O_4_) in the synthesis of alkylaminophenols at different temperatures and times. Consequently, CoFe_2_O_4_ was found to be the most suitable catalyst for the reaction due to its ease of separation by an external magnet. After the optimization of reaction conditions (15 mol% CoFe_2_O_4_, CH_3_CN, 80 °C) the effect of electron-donating and electron-withdrawing substituents on salicylaldehyde and aryl boronic acid was evaluated. In that respect, salicylaldehydes with substitutions (5-Br, 3,5-Br, 4-OMe and 5-Cl); different amines, i.e., piperazine, morpholine, pyrrolidine and *N*-dibenzylmethyl amine; and boronic acids with substitutions (4-*t*Bu, 4-Cl, 4-OMe, 3-NO_2_) were employed to obtain 72–93% yields with 19 examples. A highlighted example is presented in Figure 1. The best yield was obtained when salicylaldehyde **5** and piperidine **6** were treated with *t*-butylphenylboronic acid **7** using a CoFe_2_O_4_ catalyst in acetonitrile at 80 °C to afford alkylaminophenol **8** in 93% yield. Some significant attributes of this reaction are benign rection conditions, moderate to high yields and the short duration required for reaction completion [32].

Recently, Rafiee and Hosseinvand reported the synthesis of a nanocomposite composed of graphene oxide modified with *N*-amidinoglycine having immobilized copper ions. This nanocomposite was efficiently utilized in a Petasis–Borono Mannich reaction to synthesize aminophenol **10** with 100% conversion in a solvent-free environment via Fe_3_O_4_@GO@AG@Cu^II^ catalysis, as presented in Figure 2 [33]. The substrate scope of this methodology includes the employment of salicylaldehydes with 3-OMe, 4-OMe and 5-Br substitutions; boronic acids with 3-Me, 3-OMe and 4-acyl; and an amine component, i.e., piperidine, morpholine and 2-methyl piperidine, to provide 65–100% conversion of reagents. The salient features of this methodology were excellent catalytic activity for five consecutive runs, best yields, brief reaction times and facile recovery.

Magnetic nanoparticles, consisting of metals or their oxides, have received much attention in recent years owing to a number of mention-worthy properties, which include non-toxic nature, low cost, and oxidative stability, making them a good candidate for catalyzing several reactions, including the Petasis–Borono Mannich reaction. Shivashankar and Chacko synthesized novel aminomethylphenols bearing a piperidine moiety (80–94%) by using a catalytic amount of magnetic Fe_3_O_4_ nanoparticles. For that purpose, several substituted salicylaldehydes with 5-Br, 5-NO_2_, 5-OMe and 5-Me substitutions were reacted with boronic acids having 4-Cl and 4-Br substitution and substituted imidazole and indole to furnish 80–94% yields of target compounds. The best results were obtained when 2-hydroxy-5-methoxybenzaldehyde **11** was allowed to react with phenylboronic acid **9** and amine **12** to furnish aminomethylphenol **13** in 94% yield using 2 mol% of nano Fe_3_O_4_ in dry dioxane at room temperature. The catalyst provided high yields of target compounds even with four consecutive reruns (Figure 3) [34].

### 2.2. Petasis Reaction Involving Chiral Catalysts

The synthesis of chiral compounds and their utility has been a topic of discussion. For that purpose, chiral catalysts are employed. Han et al. synthesized alkylaminophenols (57–87% yield range and enantioselectivities 40–86% ee) via a Petasis reaction using chiral BINOL (1,1′-bi-2-naphthol) as a catalyst. Maximum yield was obtained by treating salicylaldehyde **5** with morpholine **14** and 4-methoxybenzoic acid **15** using chiral BINOL **16** (***R***) as a catalyst in mesitylene at 0 °C to obtain compound **17**(***R***) in 87% yield with 53% ee (Figure 4) [35]. Besides that, substituted aldehydes bearing (5-NO_2_, 5-OMe and 5-Bz), diverse amine components (pyrrolidine and piperidine) and different boronic acids bearing (4-OMe, 4-OBn, 4-OMOM, 4-Cl) substitutions were also employed for this reaction. Thus, asymmetric synthesis with the BINOL catalyst exhibits a wide substrate scope and acceptable enantioselectivities.

Considering the efficient activity of organocatalysts synthesized using a thiourea moiety in organic reactions, Han et al. treated a variety of salicylaldehydes with secondary amines and organoboronic acids to attain 48–92% yield range of the targeted compounds using a thiourea-derived catalyst. It was observed that 5-methoxy and 5-nitro substituted salicyladehydes; 4-OMe, 4-OBn, 4-OMOM and 4-Cl substituted boronic acids; and morpholine, piperidine, pyrrolidine and tetrahydroisoquinoline have been used in this approach efficiently. The excellent yield of alkylaminophenol **23**(***R***) (92%) with 90% ee was obtained when salicylaldehyde **5** was treated with pyrrolidine **18** and 4-methoxybenzoic acid **15** using catalyst **19**(***R***) in methyl *t*-butyl ether (MTBE) (Figure 5) [36].

Owing to the synthetic importance of amino acids (which are the building blocks of various peptides) [37], Lou and Schaus described the synthetic pathway for different amino acids (71–94%) using (*S*)-VAPOL catalyst **24** in 3Å MS/toluene mixture. These chiral catalysts behave as efficient catalysts for obtaining asymmetric products without using chiral substrates. The best results were obtained when secondary benzyl amine **22** was treated with alkenyl boronate **21** and glyoxylate **23** to afford α-amino acid (***R***,***E***)-**25** in 94% yield with 95:5 enantioselectivity (Figure 6) [38]. Moderate to good results were obtained when a number of substituted dibenzyl amines and boronates were employed.

### 2.3. Acid-Catalyzed Petasis Reaction

Reddy et al. described the lanthanum-catalyzed Petasis reaction under microwave irradiation by giving an 80–98% yield range of the desired products. The highest yield (98%) of tertiary amine **27** was recorded when aldehyde **5** was treated with morpholine **14** and (2-chlorophenyl)boronic acid **26** under microwave irradiation using La(OTf)_3_ catalyst (Figure 7). However, by using benzaldehydes (instead of salicylaldehydes) under the same reaction conditions, moderate yields of the corresponding products were observed (two examples, 75% and 78% yields) [39].

Vinylglycines or β,γ-unsaturated α-amino acids represent a unique class of optically pure nonproteinogenic amino acids, which display antimicrobial activity and act as synthetic intermediates in various organic transformations [40]. The synthesis of the aforementioned amino acids has been a tedious task. In this respect, Li and Xu in 2012 demonstrated an InBr_3_-catalyzed Petasis reaction to afford the desired amino acids within a short time in moderate to good yields (57–78%) yield range with high selectivities (up to 99%). The approach employed for that purpose includes Lewis acid as a practical and mild strategy to access functionally diverse products possessing high enantioselectivities. For example, α-(2-benzofuranyl)-glycine **31** was obtained in maximum yield (78%) with 99% diastereoselectivity through the reaction of benzofuran-2-boronic acid **28** with *N*-*t*-butanesulfinamide **29** and glyoxylic acid **30** using InBr_3_ in dichloromethane at r.t. (Figure 8) [41]. A variety of styryl substituted boronic acids with 4-Me, 4-OMe, 4-F, 4-Cl, 3-Cl, 4-CF_3_, 4-Ph and benzothiophene substitutions gave moderate yields when reacted. The target compounds obtained by this methodology have found a variety of applications in asymmetric and medicinal synthesis.

Zhang et al. developed a direct approach for the preparation of azaarene-based glycines (65–94% yield range) through a trifluoroacetic acid (TFA)-promoted Petasis reaction. Trifluoroacetic acid acted as an efficient medium for carrying out the synthesis of arylglycine derived from indoline in the presence of dichloromethane at room temperature. A highlighted pathway is presented in Figure 9, according to which the reaction of glyoxylic acid **30** with 5-nitroindoline **32** and boronic acid **33** gave indoline-derived glycine **34** in 94% yield using TFA in DCM solvent (Figure 9) [42]. The effect of various boronic acids bearing 4-methyl, 4-ethyl, 4-isopropyl, 3,5-dimethyl, 4-methoxy, 3,4,5-trimethoxy, 3,4-(1,4-dioxan-2,3-diyl), 4-SCH_3_, 4-chloro, 4-bromo, 3,5-dibromo, 4-CO_2_CH_3_, 4-CF_3_, 3,5-di-CF_3_, 4-CH_2_Br, 4-CH_2_Ph and 4-NO_2_ was also demonstrated.

The extensive utilization of organotrifluoroborate salts in palladium- and rhodium-catalyzed reactions increases the importance of these salts in organic chemistry [43]. In this regard, these borate salts were added to imine and enamine by Carrera in 2017 using a stoichiometric amount of TFA for the preparation of diversified amines (65–99%). For that purpose, a boronate complex with a catalytic amount of trifluoroacetic acid was synthesized and employed as nucleophiles. The complexes exhibit moisture and air stability and are easily available. A highlighted example is presented in Figure 10 [44]. A Petasis reaction of imine **35** with trifluoroborate **36** in the presence of acetonitrile at 23 °C afforded corresponding product **37** in 99% yield. Additionally, thiophene substituted imine **35**; the effect of 4-OMePh, 4-CF_3_Ph, 3-MePh, furan-yl, n-propyl, benzyl, heptenyl and isopropyl substitutions on the imine component were also observed.

Allylic alcohol has been broadly used to construct different types of natural products, for instance, alotaketal A and lehualides [45,46]. In this respect, Ding et al. performed a hydrochloric acid (HCl) promoted Petasis methodology for the preparation of allylic alcohols (36–98%). The use of HCl provided a smooth and efficient protocol to obtain a variety of amines having a heteroaromatic ring at adjacent positions. A highlighted example is described in Figure 11 [47]. 2-Pyridinecarbaldehyde **38**, cyclic amine **40** and oxaborole **39** were subjected to a Petasis reaction using HCl in acetonitrile at 80 °C to afford the expected allylic alcohol **41** in 98% yield. It was observed that electron-donating groups such as alkyl and aryl on an oxaborole moiety gave a good to excellent yield (75–92%), while the methoxyl group resulted in a lesser yield (60%), but the overall yield range of these groups is good in contrast to electron-withdrawing substituents. Similarly, the effect of cyclic and acyclic amines was investigated. It was observed that cyclic amines gave the Petasis products in an excellent yield (82–96%) compared to acyclic amines (48–56%).

### 2.4. Catalyst-Free Petasis Reaction

The use of transition metal-catalyzed multicomponent reactions has been a way to go for synthesizing natural products and derivatives. But these approaches pose serious hazards to the environment, causing pollution and posing difficulties in separation. These problems have been tackled by devising a catalyst-free synthesis. Considering the synthetic utility of propargylamines in natural product synthesis [48], the preparation of *N*-benzyl propargylamines (31–94%) was attained under a catalyst-free environment by Feng et al. Maximum yield (94%) was obtained when benzylamine **42** was reacted with propiolic acid **43**, boronic acid **15** and formaldehyde **44** in 1,2-dichloroethane at 45 °C (Figure 12) [49]. The benzyl amine component with 4-Cl, 2-Cl, 4-F, 4-Br, 4-CN, 2-Me, 3-Me, 4-Me, 2-OMe, 4-OMe and 3,4-OMe, along with cyclohexyl and methoxy propyl substitutions, were employed. A variety of boronic acids with different substitutions (Ph, 3-ClPh, 3-FPh, biphenyl, 3-OMePh, naphthyl, 3-thiophene, 4-FPh, 4-MePh, 2-OMePh and 4-OMePh) and propiolic acids bearing (2-FPh, 4-CF_3_Ph, 4-OMePh, biphenyl, 4-*t*BuPh, butyl, propyl and methyl) substitutions were also employed to demonstrate the substrate scope of developed procedure. This methodology accommodated a number of advantages over metal-catalyzed reactions, providing a wide substrate scope, mild reaction conditions and chemoselective products.

Antimicrobial, antiallergic, antitubercular and herbicidal properties of oxazolidines make them suitable candidates in medicinal chemistry. This significant moiety not only act as chiral auxiliaries in asymmetric synthesis but also as ligands for catalysts based on transition metal [50]. In this respect, Zheng et al. presented synthetic pathway for oxazolidines (43–85%) by the reaction of a variety of organoboronic acids with 1,2-amino alcohols and formaldehyde. For that purpose, a catalyst-free, facile and effective methodology was adopted. The maximum yield of respective product **48** (85%) was attained by the treatment of phenyl boronic acid **47** with 1,2-amino alcohol **46** and formaldehyde **44** in toluene at 80 °C (Figure 13) [51]. The usage of a number of substituted aminoethanols bearing Me and Ph as well as styryl and phenyl boronic acids having (4-OMe, 2-OMe, 4-Me, 4-Cl, 2-Me, 3-OMe, 2-F, 4-F, 4-CF_3_, 4-*t*Bu) substitutions elucidate the wide substrate scope.

Chiral α-aryl glycines have served as valuable scaffolds due to a variety of applications in the fields of pharmaceutical, biological and synthetic chemistry [52]. Considering their importance, Nanda and Trotter elaborated a Petasis–boronic acid–Mannich reaction using HFIP (hexafluoroisopropanol) as an additive in DCM (CH_2_Cl_2_) at r.t. After applying these conditions on different pyrrolidines (2-Me, 2-isopropyl, 2-phenyl, 2-methoxymethyl, 2,5-dimethyl, 3-NHBoc) and 2-methylpiperidine, glyoxylic acid monohydrates and different types of aryl boronic acids (Ph, 2-MePh, 4-OMePh, 4-FPh, 3-FPh, 3-F-4-OBnPh, 3-CNPh, thiophen-2-yl and thiophene-3-yl) furnished moderate to high yields (59–96%) of the targeted compounds. To achieve best results (96% yield), amine ***rac*-49** was treated with phenylboronic acid **9** and glyoxylic acid monohydrate **30** to obtain compound **50** with a 95:5 diastereomeric ratio (Figure 14) [53]. The main attributes of this reaction included the employment of HFIP as a co-solvent, which significantly enhanced the reaction quality by reducing the duration of the reaction.

The synthesis of natural products has been carried out via a number of strategies [54]. Natural products having quinoline rings display excellent antibacterial, antidiabetic and anti-inflammatory activities [55]. Nitrogen-containing compounds are considered significant among various natural compounds, i.e., alkaloids. In this respect, Chang et al. proposed a novel synthetic pathway for dihydroquinolines and dihydroisoquinolines. The synthesis of substituted dihydro isoquinoline was carried out by treating 2-benzofuranboronic acid with several isoquinolines. For example, the treatment of isoquinoline **51** with 2-benzofuranboronic acid **28** using DEPC (diethyl pyrocarbonate) and DCM as a solvent at r.t. gave dihydroisoquinoline **52** in 97.5% yield (Figure 15) [56]. By employing substituted quinolines with substitutions (6-yl pivalate, 3-methyl, 6-methyl, 6-yl-pivalamide and 6-nitro), substrate scalability of this method was proved and furnished high yields of respective target compounds. The main features of this methodology include simple experiments, good yields and facile reaction conditions.

Ionic liquids are efficient and unusual solvent systems introduced in recent years. These consist of almost five anions and cations. The selection of ionic liquids for some reaction depends upon the nature of reagents, the attributes of ionic liquids and the reaction conditions. These can be efficiently utilized as solvents for reaction chemistry, including catalytic reactions, due to their most attractive property, i.e., a lack of volatility [57]. Alkylaminophenols (70–85%) were prepared by Yadav via an ionic liquid-accelerated Petasis reaction involving salicylaldehydes bearing 6-methoxy and 6-ethoxy, amines (morpholine, 4-methylpiperidine, pyrrolidine, *N*-methyl-1-phenylmethanamine and *t*-butyl piperidin-3-ylcarbamate) and boronic acids (1,1-biphenyl, naphthalene, thiophene and benzodioxol-5-yl). The excellent yield of alkylaminophenol **56** (85%) was obtained via the Petasis reaction of an aldehyde **53** with cyclic amine **54** and aryl boronic acid **55** at r.t. in [Bmim]BF_4_ (Figure 16) [58]. The increased reactivity of boronic acids due to the ionic liquid led to the better yields and reduced reaction durations. The facile separation of the catalyst from the reaction mixture and its reuse makes it a good candidate as a green catalyst.

The prevalence of different forms of cancers and discovery of different compounds for their treatment have been a persistent part of research [59]. Dihydropyrone (DHP)-based natural products show antioxidant, spasmolytic, anticancer, antibacterial, antianaphylactic and anti-HIV activity [60]. A protocol for the preparation of functionalized dihydropyrones using DCM and *i*-PrOH as a reaction medium was developed by Li et al. In their methodology, 3,6-dihydropyrones were obtained in 78–98% yield, and a 45–98% yield range was observed in the case of 5,6-dihydropyrones. Amine **58** and glyoxylic acid **30** were reacted with oxaboroles **57** in DCM at 40 °C and in *i*-PrOH at 80 °C to afford 3,6-DHP **59** (98%) and 5,6-DHP **60** (98%), respectively (Figure 17) [61]. The synthesis of target compounds in high regioselectivities accounts as the prominent attributes of this methodology.

Deep eutectic solvents are unusual liquids with a bunch of useful properties, such as high chemical and thermal stability, dissolution capacity and vapor pressure. Additionally, they are environment friendly alternatives to conventional solvents owing to their reusability. Azizi et al. proposed a deep eutectic solvent-based synthesis of alkylaminophenol derivatives in good to moderate yields. The target compounds were obtained through a reaction of salicylaldehyde **5**, morpholine **14** and phenylboronic acid **9** (Figure 18) [62]. 

The aminomethyl phenol scaffold exhibits a variety of applications in agrochemicals, pharmaceuticals and plant protection. To achieve the synthesis of this moiety, Di et al. reported a catalyst-free Petasis–borono-Mannich reaction to give a variety of substituted aminomethyl phenol derivatives. The reaction was carried out in the presence of cyclopentyl methyl ether at 80 °C between substituted salicylaldehydes, secondary amines and a variety of boronic acids to furnish final compounds in low to moderate yields. The best yield was obtained when 5-bromo salicylaldehyde **62**, fused boronic acid **64** and amine **63** were made to react in the presence of cyclopentyl methyl ether (CPME) at 80 °C. The characteristic features of this reaction are green and toxic-free reaction conditions, ease of operation and the use of CPME as a solvent furnishing target compounds in good yields (Figure 19) [63].

### 2.5. Solvent-Free Synthesis via Petasis Reaction

Solvent-free protocols have gained substantial importance in organic chemistry because these reactions utilize less energy and give the desired compounds within a short time at low cost [64]. In this regard, Hosseinzadeh et al. synthesized alkyl- and arylaminophenols (83–97%) through the coupling of aryl boronic acids with alkyl amines/anilines and salicylaldehyde. For instance, treating aryl boronic acid **15** with morpholine **14** and salicylaldehyde **5** gave corresponding product **66** (97%) under ball-milling conditions. The reactivity of salicylaldehyde toward this Petasis reaction was investigated. After utilizing the same conditions and electron-donating and -withdrawing substituents on boronic acids (Ph, 3-MePh, 4-*t*BuPh, 4-ClPh) and amines (morpholine, piperidine, piperazine, dibenzyl amine, *N*-methyl benzylamine, benzyl amine, aniline, 4-aminobenzonitrile, 4-nitro aniline, 4-methoxy aniline and diphenyl amine), a 70–97% yield range was obtained (Figure 20) [65].

A new solvent-free Petasis multicomponent reaction was accomplished by Nun et al. under microwave irradiation to obtain a 44–96% yield range of the coupling products. Maximum yield (96%) was reported for fused phenylboronic acid **67** when it was treated with morpholine **14** and salicylaldehyde **5** at 120 °C (Figure 21) [66]. Various other amine substrates (morpholine, piperidine, dibenzyl amine and propenyl amine) and substituted boronic acids (Ph, 4-MePh, 3-MePh, 2-MePh, 3,4,5-OMePh, 3-NO_2_Ph, 3-OMePh, benzofuran and 5,5,8,8-tetramethyl-5,6,7,8-tetrahydronaphthalen-2-yl) were treated under microwave irradiation to obtain the coupled product. The neat conditions employed for this methodology provided the facile separation and purification of the products.

Owing to the significant position of quinoxalines in the pharmaceutical field [67,68,69,70], microwave-assisted synthesis of quinoxaline has been demonstrated by Ayaz et al. The reaction of diamines; glyoxals bearing 4-F-Ph, 4-CF_3_-Ph, Ph and 2,4,6-tri-Me-Ph; and boronic acids bearing 4-MePh, 3-FPh, 2-MeOPh, 2-naphthyl, trans-β-styryl, 2,4,6-tri-FPh and 3-CF_3_Ph furnished quinoxalines in low to excellent yields (35–98%). The reaction of diamine **69** with phenylglyoxal **70** and 2,4,6-trifluorophenylboronic acid **71** gave compound **72** in 83% yield under microwaving at 120 °C. Compound **72,** upon treatment with TFA/DCE (1,2-dichloroethane) at r.t., provided targeted quinoxaline **73** in 98% yield (Figure 22) [71].

### 2.6. Base-Catalyzed Petasis Reaction

Various synthetic strategies for 2,5-dihydrofuran have been reported because of their extensive involvement in the synthesis of natural products [72]. Cui et al. presented the amine-promoted formation of 2,5-dihydrofurans in 71–92% yield. To achieve the desired compounds, a number of amines were tried and tested to obtain the best condition, among which secondary amines bearing electron-rich alkyl groups exhibited higher yields. An exemplary substrate is presented in Figure 23 [73]. 2,5-Dihydrofuran **74** was obtained in 92% yield by the Petasis reaction of salicylaldehyde **5**, morpholine **14** and oxaborole **39** at 85 °C. A variety of substituted salicylaldehydes having 5-I, 3,5-*t*-butyl, 5-NO_2_, 5-Br, 3-Me and 3-OMe substitutions and oxaborole with phenyl, ethylbenzene, *p*-tolyl, *m*-tolyl, ethyl and butyl substitutions were also employed for this methodology.

The combination of amino acids and different polymers gives unique characteristics to the resulting compounds [74]. Takahashi and Kakuchi optimized reaction conditions to attain polymers through the coupling of aromatic boronic acids, amines and glyoxylic acids by using AIBN (2,2′-Azobis(isobutyronitrile)) in 2021. The use of derivatives of boronic acid causes some disadvantages. This problem was tackled by using MIDA esters of boronic acid, which were further polymerized to yield poly amino acids. For this purpose, boronic acid was protected with *N*-methyliminodiacetic acid (MIDA) ester. MIDA ester substituted with styrene **75** underwent free-radical polymerization in an AIBN/DMSO mixture and converted into polymer (PSt-BMIDA) **76** with 98.4% yield. Furthermore, synthesized polymer (PSt-BMIDA) **76** was subjected to a Petasis reaction with dibenzylamine **77** and glyoxylic acid **30** using a tetramethyl ammonium hydroxide (TMAOH) base in DMSO to afford the poly (α-amino acid) **78** (73.1%) (Figure 24) [75].

### 2.7. Miscellaneous Catalysts

A natural polymer, chitosan, is used in hydrogels due to the polymer’s biocompatibility, low toxicity and degradability [76]. The research group of Reddy used chitosan in a Petasis–borono-Mannich reaction for the construction of alkylaminophenols. Besides giving an excellent yield range, this catalyst can be recycled up to ten times, which highlights the significance of this method. The treatment of *ortho*-hydroxy aldehyde **5** with 4-chloro phenylboronic acid **79** and morpholine **14**, under the optimized conditions (chitosan in 1,4-dioxane at 80 °C), afforded product **80** in 95% yield, as described in Figure 25 [77]. Overall, a yield range of 86–95% was furnished when substituted 4-methoxy bearing salicylaldehyde was reacted with an amine component (morpholine, piperidine and pyrrolidine) and boronic acids with naphthyl and phenyl substitutions bearing 4-Cl, 4-OMe, 3,4-OMe, methoxycyclopropyl, 3-OMe, 2-Me, 4-Me, 3-NO_2_ and 3,4,5-OMe.

Photocatalysts are regarded as effective catalysts in radical-based reactions. Vytla et al. developed a photocatalyst ([Ir{dFCF_3_PPy}_2_(dtbpy)]PF_6_) and employed it for the synthesis of sulfonamides, amides and hydrazides via a Petasis reaction. The reaction took place between substituted aldehyde **81**, substituted amines **82** and potassium cyclohexyltrifluoroborate **83** in the presence of photocatalyst, sodium bisulfate in dichloroethane for 24 h at room temperature. This three-component method is considered efficient as it has a wide substrate scope and requires mild reaction conditions (Figure 26) [78].

## 3. Conclusions

On the basis of the literature reports discussed in this article, it is safe to conclude that the Petasis reaction is a highly efficient, atom-economical, one-pot, sustainable and waste-reducing approach. Synthetic methodologies based on the Petasis reaction have emerged as prestigious approaches to synthesize pharmaceutically relevant and structurally diverse scaffolds. This reaction proceeds well in the presence of different transition metal catalysts, Lewis acids, metal complexes bearing chiral ligands (BINOL), chitosan and ionic liquids. The functional group tolerance (substrate scope of this reaction) and diversity of the products obtained through this reaction are remarkable. Various groups have reported different alternate reaction protocols to achieve a Petasis reaction, and each methodology offers its own merits over the others, such as improved reaction yields, reduced reaction times, the recyclability of catalysts and mild reaction conditions. Albeit the extensive research conducted on Petasis reactions, the authors of this review article believe that there is still a great space to fill regarding the synthetic application of the Petasis reaction to produce natural products and drugs. Until now, only a few reports have been published involving the use of the Petasis reaction toward obtaining natural products. Moreover, the development of more and more alternate green/environmentally friendly reaction conditions for Petasis reactions are also needed at this time.

## Data Availability

All data are contained in the manuscript.

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
