# Peer review of "Recent Trends in the Petasis Reaction: A Review of Novel Catalytic Synthetic Approaches with Applications of the Petasis Reaction"

_molecules, 2023, doi:10.3390/molecules28248032_

Round 1

Reviewer 1 Report

Comments and Suggestions for Authors

This review focuses on the recent alternative applications of the Petasis reaction aimed to the improvement of the sustainability of the preparation of pharmaceutically relevant and structurally diverse scaffolds. All in all, the manuscript should be of interest for the readers of Molecules. Nonetheless, an enhancement by a deep discussion on the advantages and the drawbacks of the described procedures should be advisable.  Moreover, an accurate revision of the text is needed as highlighted in the attached pdf file.

Author Response

Dear Sir,

Thank you very much for peer reviewing our manuscript and we appreciate your complimentary recommendations as your comments have helped us significantly to improve the manuscript. We have carefully scrutinized the suggestions mentioned by our worthy reviewers and in accordance of reviewer’s comments, we have revised the manuscript. 

In general, all the recommendations and suggestions have been addressed and incorporated in the manuscript which include following.

Response to Reviewer 1 Comments

Point 1: This review focuses on the recent alternative applications of the Petasis reaction aimed to the improvement of the sustainability of the preparation of pharmaceutically relevant and structurally diverse scaffolds. All in all, the manuscript should be of interest for the readers of Molecules. Nonetheless, an enhancement by a deep discussion on the advantages and the drawbacks of the described procedures should be advisable.  Moreover, an accurate revision of the text is needed as highlighted in the attached pdf file.

Response 1: Respected reviewer, as per your point, the text has been revised carefully and mistakes have been corrected and highlighted in the manuscript. We hope that revised manuscript would be satisfying for all requirements and will be suitable for consideration for publication.

Kind Regards

Mariusz Mojzych

Reviewer 2 Report

Comments and Suggestions for Authors

Although the information provided by the authors are intriguing, the flow of the text seems rather inconvenient. The article has many typos and omissions.Few notes on this issue:

1. There is no reference in the text to Figure 1 (page 2).

2. Why write substituent R5 in Figure 1 if these are only hydrogen atoms?

3. Figure 2 should show the key synthesis steps for compounds 1-4 in which the Petasis reaction occurs.

4. In the introduction section, authors should add a reference to a large-scale review of the Petasis reaction:

In the introduction section, authors should add a reference to a large-scale review of the Petasis reaction: https://doi.org/10.1021/acs.chemrev.9b00214

In addition, the authors pay rather little attention to the possibilities of modifying heterocyclic compounds using the Petasis reaction. In particular, it is necessary to add literature data on the modification of pyrazine derivatives, which have a wide range of biological activity: http://dx.doi.org/10.3998/ark.5550190.p008.690, https://doi.org/10.1016/j.bmcl.2014.12.025; etc.

5. The word "Alkylamine Phenols" was repeated twice (page 3, line 84).

6. An unfinished sentence that makes no sense "The synthesized catalyst." (page 4, line 116).

7. On page 5, it is not clear what configuration was the initial chiral BINOL (R- or S-) and what is the configuration of the product?

8. On page 6, the stereochemical configuration of chiral product 28 is also not clear.

9. On page 7 (line 208), there is a typo. It should be Scheme 9.

10. On page 8 (line 237), there is a typo. It should be Scheme 11.

11. On page 10 (line 302) it is stated that aniline 57 is chiral. What is its stereochemical configuration, as well as that of product 58? Appropriate changes must be made to Scheme 16.

12. The authors need to decipher all the abbreviations given in the reaction schemes. In addition, all abbreviations in the text must also be deciphered the first time they are mentioned.

13. The most important note is the complex structure of the review with many small subsections. Essentially, all the types of Petasis reactions presented can be divided into three large chapters: non-catalyzed reactions, acid-catalyzed and base-catalyzed reactions. At the same time, I would like the author not to simply re-compile one example with the highest yields, but to describe all possible combinations in the starting compounds and the resulting products. In this case, it is urgently necessary to explain

14. Finally, it is recommended to add a discussion of future directions and emerging trends in the Petasis reaction. I would like to know the opinion of the authors about what they consider the most promising in this area of chemistry.

Comments on the Quality of English Language

Moderate editing of English language required.

Author Response

Thank you very much for peer reviewing our manuscript and we appreciate your complimentary recommendations as your comments have helped us significantly to improve the manuscript. We have carefully scrutinized the suggestions mentioned by our worthy reviewers and in accordance with the reviewer’s comments, we have revised the manuscript. 

In general, all the recommendations and suggestions have been addressed and incorporated in the manuscript .

Round 2

Reviewer 1 Report

Comments and Suggestions for Authors

The revised manuscript should be suitable for publication.

Reviewer 2 Report

Comments and Suggestions for Authors

The authors have made an efficient revision on their work, which can be accepted in the current form.

Comments on the Quality of English Language

Minor editing of English language required.